# DISCOVERING LONG-TERM EFFECTS ON PARAMETER EFFICIENT FINE-TUNING

## ABSTRACT

Pre-trained Artificial Neural Networks (ANNs) demonstrate robust pattern recognition abilities, closely mirroring the functionality of Biological Neural Networks (BNNs). We are particularly intrigued by these models' capacity for acquiring new knowledge through fine-tuning, such as Parameter-efficient Fine-tuning (PEFT). Given that both ANNs and BNNs propagate information layer-by-layer, a useful analogy can be drawn: ANN weights correspond to synapses in BNNs, while features (latent variables or activations) parallel the neurotransmitters released by neurons. Building upon this clue, we delve deeper into exploring the connections between feature adjustment and weight adjustment, resulting in our proposed method Synapses & Neurons (SAN) that learns scaling matrices for features and propagates their effects towards posterior weight matrices. Our approach draws strong inspiration from well-known neuroscience phenomena - Long-term Potentiation (LTP) and Long-term Depression (LTD), revealing the relationship between synapse development and neurotransmitter release levels. We conducted extensive comparisons of PEFT on 26 datasets using attention-based networks as well as convolution-based networks, leading to significant improvements compared to other tuning methods, +8.5% over fully-finetune, +7% over Visual Prompt Tuning, and +3.2% over Low-Rank Adapter.

## 1 INTRODUCTION

The use of large pre-trained models demonstrates their robust adaptability to various downstream datasets and tasks through fine-tuning techniques. However, performing full fine-tuning by adjusting all parameters imposes significant computation and data costs. In this context, the concept of parameter-efficient fine-tuning (PEFT) aims to reduce the number of adjustable parameters during fine-tuning, resulting in fewer parameters, faster training speed, and a lower risk of overfitting Ding et al. (2023). Adhering to this rule-of-thumb strategy can simultaneously alleviate the cost burdens.

Compared to full fine-tuning, in the early stage of PEFT history, most methods were referred to as "partial fine-tuning" and focused on releasing a subset of parameters for adjustment. For example, linear probing was the simplest method that only released the head parameters of the model for fine-tuning. More advanced approaches like Bitfit (i.e., bias tuning) (Zaken et al., 2021) chose to release biases and achieved better global adjustment capabilities. Recent concepts such as "sparse training" introduced more sophisticated selection mechanisms by utilizing gradients or parameter magnitudes (He et al., 2023). However, these methods still faced challenges where the subset of parameters lacked representative abilities of the entire model. Consequently, researchers shifted their focus towards directly adjusting the output features of each layer. The most popular approach in this regard is the Low-rank Adapter (LoRA) family (Hu et al., 2021), which employs two low-rank sequential learnable matrices (down and up) alongside each layer to simulate additional parameter activities while handling input with original parameters; thus resulting in final features being a summation of LoRA's output and original output. The success of LoRA indicated that adjusting features would be more efficient and implementation-friendly. Other notable works in feature adjustment include Visual Prompt Tuning (VPT) (Liu et al., 2021a; Jia et al., 2022), where extra learnable tokens are concatenated with each layer's feature, and Scale and Shift Features (SSF) (Lian et al., 2022), which applies a linear transformation to features using learnable shift and scale factors.

From a mathematical perspective, we can consider feature transformations as approximation of parameters tuning. Operations such as addition, concatenation, and linear transformation on features essentially perform unified transformations on the weighted sum results of each channel in the parameter matrices. This viewpoint provides a more generalized framework for understanding PEFT methods. For instance, let $W$ be the original parameter matrix of a layer, and $x$ be the input feature. The output feature $y$ is typically computed as $y = Wx$. In PEFT methods that transform features:

- Addition (as in LoRA):
  - $y' = Wx + Ax$, where $A$ is a low-rank matrix and can be viewed as a transformation on $W$: $W' = W + A$.
- Concatenation (as in VPT):
  - $y' = W[x; p]$, where $p$ is the prompt vector and effectively extends $W$ to $W' = [W, W_p]$, where $W_p$ is the weight for the prompt.
- Linear transformation (as in SSF):
  - $y' = \alpha \odot (Wx) + \beta$, where $\alpha$ and $\beta$ are learnable scale and shift factors and can be seen as transforming $W$ to $W' = \alpha W$, with an additional bias term.

These feature-level operations can be interpreted as implicit transformations of the entire parameter space, offering a more flexible and efficient way to adapt the model Zhang et al. (2024). By operating on features, we're essentially performing a form of meta-learning, where the **model learns how to model its original parameters indirectly through the additional parameters** created for feature modifications.

The success of feature-based PEFT methods raises an intriguing question: *Should those additional parameters created for feature modifications in one layer affect the parameters of subsequent layers?* We found these aforementioned methods overlooked this and focused only on the bandwidth of the current layer, yet it finds significant resonance in neuroscience, particularly in the phenomena of Long-Term Depression (LTD) and Long-Term Potentiation (LTP). In neuroscience, LTD and LTP are well-established mechanisms of synaptic plasticity that play crucial roles in learning and memory formation Bliss & Collingridge (1993); Malenka & Bear (2004). LTP refers to the strengthening of synaptic connections, while LTD refers to their weakening. Specific patterns of short-term neural activity typically induce these processes and can persist for extended periods, hence the term "long-term" Citri & Malenka (2008).

A key aspect of LTD and LTP is their ability to induce changes in the immediate synaptic connection and subsequent neurons along the pathway, as shown in Figure 1. For instance, studies have shown that by modulating the neurotransmitter release levels of presynaptic neurons (often through pharmacological or optogenetic methods), researchers can observe changes in the synaptic development of downstream neurons Takeuchi et al. (2014); Nabavi et al. (2014). This trans-synaptic effect, known as **Heterosynaptic Plasticity**, suggests that local changes can propagate and influence broader neural networks Chistiakova et al. (2014).

Drawing an analogy between neural networks and biological neural systems, we can consider features analogous to neurotransmitters and parameter matrices as synapses. Following this, a natural extension of current PEFT methods would be to allow feature adjustments in one layer to propagate and influence the parameters of subsequent layers explicitly. As a brief introduction to our proposed Synapses & Neurons (SAN) method, for each layer, we first conduct trainable scaling for each feature to mimic the rapid change in pre-synaptic neuron's neurotransmitter level when exposed to a stimulant. Further on, we propagate those scaling factors to the next layer's parameters. This simulates the further effect of post-synaptic neuron development i.e. Heterosynaptic Plasticity. Notice the trainable parameters for SAN are the scaling factors with the exact shape as the feature size, this is very efficient and can be considered as a degraded LoRA with a bottleneck i.e. rank size equal to one. However, we conducted extensive experiments on various datasets and outperformed LoRA, as well as many other methods including the existing state-of-the-art method, by a large margin.

In a nutshell, our contributions to this paper are as follows:

- **Discover deficiencies from PEFT's line of research:** We dive deep into the progress of PEFT research from adjusting parameter weights to transforming layer output features.

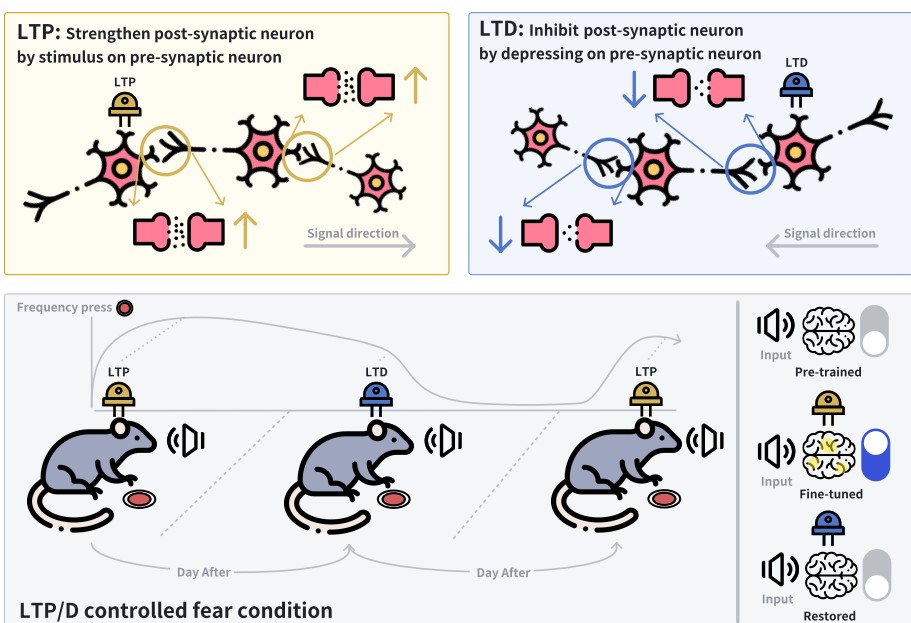

Figure 1: **The mechanisms and effects of Long-Term Potentiation (LTP) and Long-Term Depression (LTD) on synaptic transmission and behavioral conditioning: Top Panel:** LTP is depicted as the strengthening of the post-synaptic neuron in response to stimulation of the pre-synaptic neuron. This is represented by an increase in synaptic efficacy and enhanced signal transmission along the synapse and vice versa for LTD. **Bottom Panel:** Illustrates an experiment of LTP and LTD in a controlled fear conditioning paradigm using rodents, where they are utilized to either enhance or diminish the conditioned fear response.

However, most existing PEFT methods focus solely on the impact of additional parameters on the current layer, without fully addressing their influence on subsequent layers.

- **Proposed SAN method inspired by BNNs:** Inspired by the human brain and the phenomena of LTD and LTP in BNNs. We formulate the Synapses & Neurons, the first PEFT method that allows feature adjustments in one layer to propagate and influence the parameters of subsequent layers explicitly.

- **Theoretical proofs and extensive experiments verified our method to outperform SOTA:** We demonstrate the effectiveness of our approach through theoretical proofs and extensive experiments on multiple benchmarks, including FGVC, VTAB-1k, and General Image Classification. Our method consistently surpasses the current state-of-the-art by $1\% \sim 4\%$.

## 2 RELATED WORKS

**Parameter-efficient fine-tuning**     The most straightforward method for PEFT is linear probing Alain & Bengio (2016). By simply adding or modifying a trainable new head, the pre-trained model can adapt to new tasks. However, the expressiveness of the linear probing method is limited. An intuitive improvement proposed in Bitfit Zaken et al. (2021) involves unfreezing the bias. A more efficient approach is prompt tuning Jia et al. (2022); Liu et al. (2021a), which adjusts the inputs. Recent work, such as sensitivity-aware PEFT, analyzes weight magnitudes or accumulated gradients on specific datasets to discover model sparsity and focuses only on tuning those areas. In addition to focusing on adjustment locations, there are also differences in how adjustments are made. The adapter Zhang et al. (2020) is a commonly used approach that adds an extra layer along or inserts it into the pre-trained model; features pass through this adapter and output an adjustment value for different adjustment operations (etc. addition, production, and concatenation). Another efficient adapter style is Low-Rank

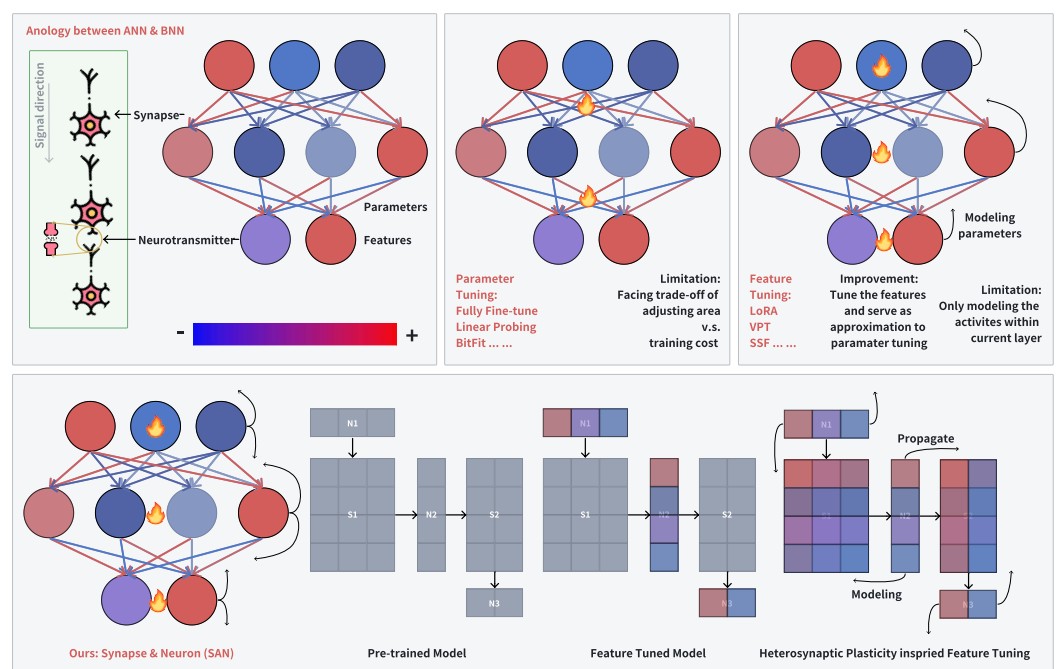

Figure 2: **Synapses & Neurons (SAN) pipeline: Top left:** Analogy between ANNs and BNNs, illustrating the concept of Synapse & Neuron (SAN) tuning. The figure compares different fine-tuning approaches: **Top middle:** Traditional parameter tuning methods like fully fine-tuning and linear probing. **Top right:** Feature tuning methods such as LoRA, VPT, and SSF. **Bottom:** Our proposed SAN method demonstrates how SAN incorporates both synapse (parameter) and neuron (feature) tuning by modeling backward parameters and propagating to forward parameters, inspired by Heterosynaptic Plasticity, to achieve more effective and efficient transfer learning.

Adapter (LoRA) Hu et al. (2021), which uses two low-rank matrices to equalize a dense layer. This adapter style can be further quantized into Q-LoRA Dettmers et al. (2024) and other types of matrix decompositions. Reparameterization offers a completely different style of adjustment by directly shifting and scaling features; this remapping technique is known as Shift and Scale PEFT Lian et al. (2022).

**Long-Term Depression/Potentiation and Heterosynaptic Plasticity**  Long-Term Depression (LTD) and Long-Term Potentiation (LTP) are fundamental mechanisms of synaptic plasticity in biological neural networks, playing crucial roles in learning and memory formation Malenka & Bear (2004). LTP refers to the strengthening of synaptic connections, while LTD involves their weakening, both persisting for extended periods Bear et al. (2007). These processes are primarily triggered by specific patterns of neuronal activity and are often considered Hebbian in nature, following the principle that "neurons that fire together, wire together" Hebb (1949). However, the discovery of heterosynaptic plasticity has expanded our understanding of synaptic modulation beyond this simple associative rule Chistiakova et al. (2014). Heterosynaptic plasticity refers to changes in synaptic strength that occur at synapses that are not directly involved in the inducing activity Bailey et al. (2000). This form of plasticity allows for more complex and distributed forms of information storage and processing in neural networks Lynch (2004). For instance, when LTP is induced at one set of synapses, heterosynaptic LTD may occur at nearby inactive synapses, potentially serving as a homeostatic mechanism to maintain overall network stability Royer & Paré (2003). The molecular mechanisms underlying these forms of plasticity involve complex cascades of intracellular signaling. Heterosynaptic plasticity, involves unique signaling pathways, including the spread of intracellular messengers and the release of diffusible factors that can affect nearby synapses Oh et al. (2015).

## 3 METHODS

### 3.1 PRELIMINARIES

**Transformers:** For vision transformers (VIT), RGB input image with shape $I \in R^{3 \times H \times W}$ is divided into $N \times N$ patches. A convolution layer is used to convert patches to embeddings, with an extra class token appended to the end. The input for transformer blocks is $x \in R^{(N^2+1) \times d}$, where $d$ is the dimension for each embedding. These embeddings use self-attention algorithms to calculate the dependencies.

The attention mechanism is defined as:

$$\text{Attention}(Q, K, V) = \text{Softmax}\left(\frac{QK^T}{\sqrt{d}}\right) V \tag{1}$$

where queries, keys, and values $Q, K, V \in R^{(N^2+1) \times d}$.

**LoRA & Adapter:** These PEFT methods use two low-rank learnable matrices (Down and Up) to simulate the full rank dense layers.

$$\text{out} = [W^{up}\phi(W^{down}x^T)]^T, \tag{2}$$

where $W^{down} \in R^{d' \times d}(d' \ll d)$, $\phi$, and $W^{up} \in R^{d' \times d}$ represent the down-projection matrix, non-linear function, and up-projection matrix, respectively.

**Visual Prompt Tuning:** This PEFT method concatenates a learnable prompt $p \in R^{n \times d}$ to each input $x$, resulting in $x' = [x; p] \in R^{(N^2+n+1) \times d}$.

**Scale & Shift Features:** This PEFT method applies a learnable linear transformation to each layer's output $y' = \gamma \odot y + \beta \in R^{(N^2+1) \times d}$, where $\gamma, \beta \in R^d$ are the scaling and shifting factors, respectively, and $\odot$ is the element-wise product. The reparameterize formula is:

$$y' = \gamma \odot y + \beta = \gamma \odot (w * x + b) + \beta = (\gamma \odot w) * x + \gamma \odot b + \beta, \tag{3}$$

where $b$, $w$, and $x$ are bias, weight, and input for this layer. $*$ is convolution or multiplication operation in convolution or MLP layer. SSF is indeed highly efficient, however, this method, which we considered, ignores the effect of pre-synaptic stimulus and causes post-synaptic development, so our major modification would be conducted on it.

### 3.2 SAN: SYNAPSES AND NEURONS

#### 3.2.1 BASIC FORMULA:

Similar to SSF, the scaled output $y'^l$ of layer $l$ can be described as $y'^l = \gamma^l \odot y^l$, where $\gamma^l$ is the scaling factor (we initialize it to one) of our SAN adapter and $y^l$ is the original output of the linear transformation. Then, the output goes through a set of operations such as activation function or normalization, denoted $\sigma(\cdot)$. We consider the scaling of output in this layer, i.e., $\gamma^l$ would pose a further effect towards the next layer's parameters (similar to the LTD/P effect found in BNNs), so we further apply it to scale the parameters in the next layer $w^{l+1}$. The parameters for the next layer would be scaled to $w'^{l+1} = \gamma^l \odot w^{l+1}$; therefore the output for the next layer is:

$$y'^{l+1} = \gamma^{l+1} \odot \left(w'^{l+1} * \sigma(y'^l) + b^{l+1}\right), \tag{4}$$

where $b^{l+1}$ is the bias of the next layer. Our SAN pipeline is depicted in Figure 2.

#### 3.2.2 REPARAMETERIZATION:

As introduced in Eq. 3, the reparameterization of SSF implies a strong assumption: for each row of the current layer's weight matrix, the scaling & shifting factors would be the same. In contrast, our SAN method introduces a critical re-application process of the scaling factor to the next layer's weight matrix. This approach allows us to achieve a unique adjustment value for every individual parameter without incurring any extra training burden.

By propagating the scaling factor $\gamma^{l-1}$ from the previous layer to the current layer's weight, we can overcome the strong assumption of SSF and achieve a more fine-grained adjustment. The reparameterization formula of SAN can be expressed as:

$$
\begin{aligned}
y'^l &= \gamma^l \odot (\gamma^{l-1} \odot w^l * x + b^l) + \beta^l \\
&= (\gamma^l \odot \gamma^{l-1} \odot w^l) * x + \gamma^l \odot b^l + \beta^l,
\end{aligned}
\tag{5}
$$

where $\gamma^{l-1}$ is the scaling factor from the previous layer, $\gamma^l$ and $\beta^l$ are the scaling and shifting factors for the current layer, $w^l$ and $b^l$ are the weight and bias of the current layer, and $x$ is the input.

### 3.2.3 REGULARIZATION:

The re-application of scaling factors in SAN not only provides fine-grained parameter adjustment but also introduces an implicit regularization effect to prevent overfitting. This regularization emerges from the approximate quadratic nature of the scaling factor's influence when propagated through layers. To illustrate this, let's consider a simplified two-layer linear network scenario without any activation and normalization:

$$
y'^{l+1} = \gamma^{l+1} \odot ((\gamma^l \odot w^{l+1}) * (\gamma^l \odot x^{l+1}) + b^{l+1})
\tag{6}
$$

Rearranging this equation, we get:

$$
y'^{l+1} = (\gamma^{l+1} \odot \gamma^l \odot \gamma^l \odot w^{l+1}) * x^{l+1} + \gamma^{l+1} \odot b^{l+1}
\tag{7}
$$

The presence of $(\gamma^l)^2$ in this formulation reveals a crucial property: the effect of the scaling factor is essentially squared when propagated through layers. This quadratic influence acts as a soft constraint on the magnitude of $\gamma^l$, discouraging extreme values and promoting stability. To formalize this regularization effect, we can express it as an implicit regularization term $R(\gamma)$ added to the loss function:

$$
R(\gamma) = \lambda \sum_l \|\gamma^l - 1\|^2
\tag{8}
$$

where $\lambda$ is a hyperparameter controlling the strength of regularization, this regularization term penalizes large deviations of $\gamma^l$ from its initial value of 1, effectively limiting the model's capacity to make extreme adaptations.

### 3.2.4 EXPLICIT PROPAGATION:

The key innovation of our SAN method lies in explicitly propagating the scaling factor of the current layer to the parameters of the subsequent layer. This approach is motivated by a fundamental insight into the nature of linear transformations in neural networks: any linear transformation applied to features for Parameter-Efficient Fine-Tuning (PEFT) implicitly affects the subsequent layer's parameters.

To elaborate, consider a linear transformation $T$ applied to the features $f$ of layer $l$, $f' = T(f)$. The output of the subsequent layer $l+1$ with weight $W$ and bias $b$ can be expressed as, $y = W \cdot T(f) + b$. Due to the linearity of the operations, this is equivalent to:

$$
y = (W \cdot T) \cdot f + b = W' \cdot f + b
\tag{9}
$$

where $W' = W \cdot T$ is an adjusted weight matrix.

This equivalence reveals that any linear transformation of features in layer $l$ can be equalized as an adjustment to the weights of layer $l+1$, assuming no non-linear activations are applied between these operations. In essence, methods that apply linear transformations to features are implicitly learning an adjustment matrix for the subsequent layer's weights.

While this principle is straightforward in purely linear scenarios, real-world neural networks incorporate non-linear activations and normalization layers. However, we argue that our approach remains approximately valid even in these more complex settings. This approximation is based on two key observations:

1. **Near-linear behavior of modern activation functions:** Many popular activation functions, such as ReLU and its variants, exhibit approximately linear behavior in certain regions. This near-linearity allows our linear transformation principle to hold to a good approximation over significant portions of the input space.

2. **Adaptive re-calibration of scaling factors:** To account for the effects of non-linearities and normalization, SAN introduces an additional learnable linear transformation before re-applying the scaling factor to the next layer's weights. This can be expressed as:

$$\gamma'^l = A^l \gamma^l + b^l \tag{10}$$

where $\gamma'^l$ is the recalibrated scaling factor, and $A^l$ and $b^l$ are learnable parameters. The weight adjustment for the next layer then becomes:

$$W'^{l+1} = W^{l+1} \odot \gamma'^l \tag{11}$$

This adaptive re-calibration allows SAN to:

- Compensate for the non-linear effects introduced by activation functions and normalization layers.
- Fine-tune the propagation of scaling factors to better suit the specific characteristics of each layer and the task.
- Maintain the benefits of weight adjustment while adapting to the complexities of modern neural architectures.

## 4 EXPERIMENTAL EVALUATION

To assess the efficacy of our proposed SAN, we conducted extensive experiments across a diverse range of visual datasets. This section outlines our experimental framework, including the datasets utilized, the backbone architectures employed, and the baseline methods we compared against. We then present our findings, demonstrating SAN's performance and versatility. Additionally, we provide an in-depth analysis of various scaling strategies and their impacts through comprehensive ablation studies.

### 4.1 EXPERIMENTAL FRAMEWORK

**Dataset Selection** Our evaluation leverages a variety of datasets, categorized into three distinct groups:

- **Fine-Grained Visual Classification (FGVC):** This category comprises five specialized tasks, utilizing datasets such as CUB-200-2011 Wah et al. (2011), NABirds Van Horn et al. (2015), Oxford Flowers Nilsback & Zisserman (2008), Stanford Dogs Khosla et al. (2011), and Stanford Cars Krause et al. (2013).
- **Visual Task Adaptation Benchmark (VTAB-1k):** Zhai et al. (2019) This benchmark encompasses 19 diverse visual classification tasks, organized into Natural, Specialized, and Structured subsets.
- **General Image Classification:** We include CIFAR-100 Krizhevsky et al. (2009) and ImageNet-1k Deng et al. (2009), two representative benchmarks in the field.

**Model Architectures** To ensure a fair comparison with existing methods, our primary experiments employ ViT-B/16 Dosovitskiy et al. (2020), pre-trained on ImageNet-21K Deng et al. (2009).

To further demonstrate SAN's adaptability, we extend our experiments to include Swin Transformer (Swin-B) Liu et al. (2021b) and ConvNeXt-B Liu et al. (2022), representing state-of-the-art Transformer-based and CNN-based architectures, respectively.

**Comparative Methods**     We evaluate SAN against a spectrum of fine-tuning approaches, broadly classified into three categories:

- **Full Model Tuning:** This conventional approach involves updating all model parameters during the fine-tuning process.

- **Parameter Tuning Methods:** These techniques fine-tune a subset of the original model's parameters. Examples include linear probing and Bias tuning Zaken et al. (2021). While computationally efficient, these methods have historically shown limited effectiveness. Our proposed SAN falls within this category, aiming to overcome previous limitations while maintaining efficiency and broad applicability.

- **Feature Tuning Methods:** These methods introduce additional trainable parameters to the model, such as Adapter Zhang et al. (2020) and Visual Prompt Tuning (VPT) Jia et al. (2022). While effective, they often incur extra computational costs during both training and inference. Some variants, like LoRA Hu et al. (2021) and SSF Lian et al. (2022), allow for parameter reparameterization, potentially mitigating inference-time overhead.

**Implementation Specifics**     Our image processing pipeline follows the protocol established by SSF for the FGVC, VTAB-1k, and CIFAR-100 datasets. We optimize our models using Adam/AdamW Kingma & Ba (2014) with a cosine learning rate decay schedule over 100 epochs, incorporating a linear warm-up phase for the initial 10 epochs. All experiments were conducted using a distributed setup across four NVIDIA RTX 3090 GPUs to ensure the timely completion of our comprehensive study.

### 4.2 PERFORMANCE WITH VISION TRANSFORMER AS BACKBONE

Tab 1 presents a comprehensive comparison of our proposed SAN method against other state-of-the-art fine-tuning approaches using Vision Transformer (ViT-B) as the backbone. The results clearly demonstrate the effectiveness and efficiency of SAN across a wide range of tasks and datasets.

One of the striking aspects of SAN's performance is its parameter efficiency. While LoRA, we maximum its bottleneck dimension around the 1% constraint and serves as a strong baseline, SAN achieves superior performance using only 0.20% of the parameters. This parameter efficiency is comparable with SSF since re-apply operations do not introduce extra burdens. Nevertheless, SAN shows remarkable improvements over its competitors, even in challenging subsets of the VTAB dataset such as Specialized and Structure.

SAN also shows consistent performance across diverse datasets - from FGVC which focuses on fine-grained classification tasks with moderate training images to VTAB-1k which focuses on challenging varieties of subset and limited training images and more general image classification tasks like CIFAR100 and ImageNet-1k with sufficient training images - underscores its versatility and robustness. Notably, SAN outperforms full fine-tuning in many cases, despite using only a fraction of the parameters, we believe the key is SAN have a great balance between expressiveness and preventing overfitting.

### 4.3 PERFORMANCE WITH DIFFERENT BACKBONES

To demonstrate the versatility of our SAN method, we conducted experiments using three different backbone architectures: Vision Transformer (ViT-B), Swin Transformer (SWIN-B), and ConvNeXt (ConvNeXt-B). Figure 3a illustrates the performance of various fine-tuning methods across these backbone architectures.

As evident from the radar chart, SAN consistently outperforms other fine-tuning methods across all three backbone architectures. This performance consistency demonstrates the robustness and adaptability of our proposed method. The chart also reveals interesting patterns in the performance of different methods. For instance, while LoRA shows competitive performance with transformer-based

Table 1: Comprehensive performance comparison using Imagenet-21k pretrained ViT-B as backbone. Results show accuracy (%) for various fine-tuning methods across different datasets. Best results are highlighted in red (1st) and blue (2nd).

| Dataset | Linear Probing | Bitfit | LoRA | Adapter | VPT-S | VPT-D | Fully FT | SSF | SAN (Ours) |
|---|---|---|---|---|---|---|---|---|---|
| **Overall Mean Performance** | | | | | | | | | |
| Mean Param.% ↓ | 0.11% | 0.17% | 0.89% | 0.38% | 0.22% | 0.81% | 100.00% | 0.34% | 0.34% |
| Mean Acc.% ↑ | 60.55% | 68.60% | 76.12% | 63.86% | 70.31% | 74.62% | 71.76% | 77.68% | 79.26% |
| **FGVC** | | | | | | | | | |
| Mean Param.% ↓ | 0.21% | 0.33% | 0.90% | 0.48% | 0.29% | 0.99% | 100.00% | 0.45% | 0.45% |
| Mean Acc.% ↑ | 79.32% | 85.66% | 84.66% | 84.78% | 84.66% | 89.10% | 88.54% | 90.72% | 91.62% |
| CUB-2011 | 85.30% | 87.10% | 86.70% | 85.60% | 86.70% | 88.50% | 87.30% | 89.50% | 90.60% |
| NA-Brids | 75.90% | 84.30% | 78.80% | 79.80% | 78.80% | 84.20% | 82.70% | 85.70% | 86.30% |
| Oxford Flowers | 97.90% | 98.50% | 98.40% | 98.90% | 98.40% | 99.00% | 98.80% | 99.60% | 99.70% |
| Stanford Dogs | 86.20% | 89.80% | 90.70% | 87.60% | 90.70% | 90.20% | 89.40% | 89.60% | 91.10% |
| Stanford Cars | 51.30% | 68.60% | 68.70% | 72.00% | 68.70% | 83.60% | 84.50% | 89.20% | 90.40% |
| **VTAB-1k** | | | | | | | | | |
| Mean Param.% ↓ | 0.05% | 0.16% | 0.90% | 0.31% | 0.13% | 0.70% | 100.00% | 0.28% | 0.28% |
| Mean Acc.% ↑ | 53.30% | 62.06% | 72.63% | 55.82% | 64.85% | 69.43% | 65.56% | 73.10% | 75.00% |
| *Natural* | | | | | | | | | |
| Mean Acc.% ↑ | 69.09% | 73.31% | 79.76% | 70.50% | 76.81% | 78.49% | 75.99% | 81.57% | 83.19% |
| CIFAR100 | 63.40% | 72.80% | 68.10% | 74.10% | 77.70% | 78.80% | 68.90% | 69.00% | 74.30% |
| Caltech101 | 85.00% | 87.00% | 91.40% | 86.10% | 86.90% | 90.80% | 87.70% | 92.60% | 93.75% |
| DTD | 64.10% | 59.20% | 69.80% | 63.20% | 62.60% | 65.80% | 64.30% | 75.10% | 76.40% |
| Flowers102 | 97.20% | 97.50% | 99.00% | 97.70% | 97.50% | 98.00% | 97.90% | 99.40% | 99.70% |
| Pets | 86.30% | 85.30% | 90.50% | 87.00% | 87.30% | 88.30% | 86.90% | 91.80% | 93.00% |
| SVHN | 36.60% | 60.00% | 86.40% | 34.60% | 74.50% | 78.10% | 87.40% | 90.20% | 91.80% |
| Sun397 | 51.00% | 51.40% | 53.10% | 50.80% | 51.20% | 49.60% | 38.80% | 52.90% | 53.40% |
| *Specialized* | | | | | | | | | |
| Mean Acc.% ↑ | 26.85% | 44.10% | 60.23% | 32.39% | 46.98% | 55.00% | 47.64% | 58.96% | 61.00% |
| Patch Camelyon | 78.50% | 78.70% | 85.10% | 76.30% | 78.20% | 81.80% | 78.90% | 87.40% | 88.10% |
| EuroSAT | 87.50% | 91.60% | 95.80% | 88.00% | 92.00% | 96.10% | 95.70% | 95.90% | 97.70% |
| Resisc45 | 68.60% | 73.00% | 84.70% | 73.10% | 75.60% | 83.40% | 84.20% | 87.40% | 90.60% |
| Retinopathy | 74.00% | 69.80% | 74.20% | 70.50% | 72.90% | 68.40% | 73.90% | 75.50% | 78.10% |
| *Structure* | | | | | | | | | |
| Mean Acc.% ↑ | 77.15% | 78.28% | 84.95% | 76.98% | 79.68% | 82.43% | 83.18% | 86.55% | 88.63% |
| Clevr/count | 34.30% | 61.50% | 83.00% | 45.70% | 50.50% | 68.50% | 56.30% | 75.90% | 82.40% |
| Clevr/distance | 30.60% | 55.60% | 66.90% | 37.40% | 58.60% | 60.00% | 58.60% | 62.30% | 61.40% |
| DMLab | 33.20% | 32.40% | 50.40% | 31.20% | 40.50% | 46.50% | 41.70% | 53.30% | 54.50% |
| KITTI/distance | 55.40% | 55.90% | 81.40% | 53.20% | 67.10% | 72.80% | 65.50% | 80.60% | 82.10% |
| dSprites/loc | 12.50% | 66.60% | 80.20% | 30.30% | 68.70% | 73.60% | 57.50% | 77.30% | 81.70% |
| dSprites/ori | 20.00% | 40.00% | 46.60% | 25.40% | 36.10% | 47.90% | 46.70% | 54.90% | 55.21% |
| SmallNORB/azi | 9.60% | 15.70% | 32.20% | 13.80% | 20.20% | 32.90% | 25.70% | 29.50% | 30.30% |
| SmallNORB/ele | 19.20% | 25.10% | 41.10% | 22.10% | 34.10% | 37.80% | 29.10% | 37.90% | 40.40% |
| **General** | | | | | | | | | |
| Mean Param.% ↓ | 0.48% | 0.61% | 1.18% | 0.8% | 0.91% | 1.42% | 100.00% | 0.69% | 0.69% |
| Mean Acc.% ↑ | 85.37% | 88.07% | 87.95% | 88.03% | 86.23% | 87.81% | 88.70% | 88.55% | 88.90% |
| CIFAR100 | 88.70% | 93.39% | 93.53% | 93.34% | 90.38% | 93.17% | 93.82% | 93.99% | 94.11% |
| Imagenet-1k | 82.04% | 82.74% | 82.36% | 82.72% | 82.08% | 82.45% | 83.58% | 83.10% | 83.69% |

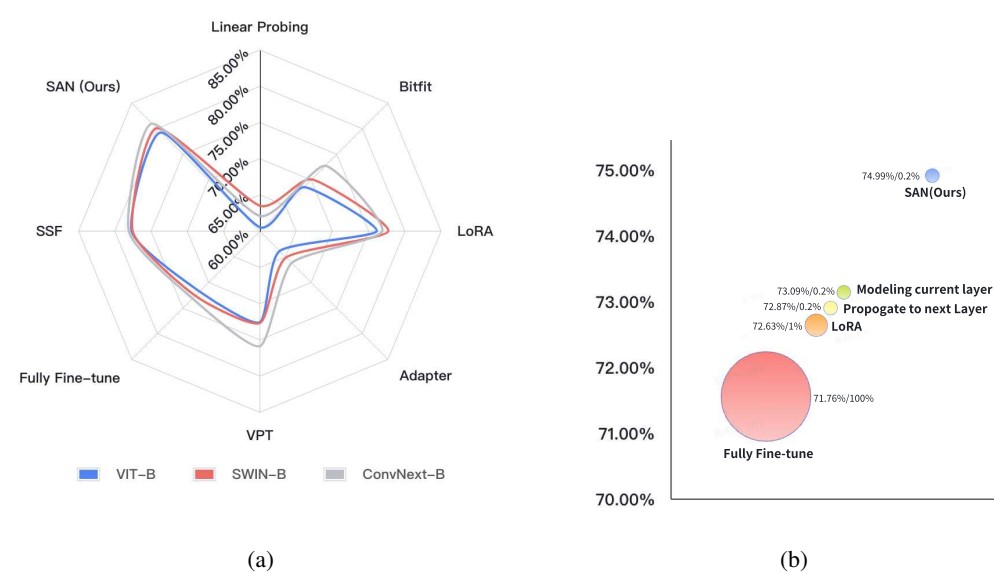

(a)                                                      (b)

Figure 3: **(a)** Performance comparison of different fine-tuning methods (VPT here use the average accuracy of VPT-shallow & deep) across various backbone architectures. The radar chart shows the mean accuracy (%) for each method using ViT-B, SWIN-B, and ConvNeXt-B as backbones. **(b)** Performance comparison of different fine-tuning methods, highlighting the contributions of modeling and propagation in SAN. The y-axis shows the mean accuracy over the VTAB-1k dataset using pre-trained VIT-B. Circle size represents the amount of trainable parameters.

models, its effectiveness slightly diminishes with the ConvNeXt-B architecture. In contrast, SAN maintains its leading position across all backbones, suggesting a more generalized approach to parameter-efficient fine-tuning.

### 4.4 ABLATION STUDIES

To investigate the effectiveness of our proposed SAN method, we conduct ablation studies focusing on separating two key components: modeling of the current layer by a set of learnable scaling factors and propagation of the learned scaling factors to the next layer. These studies aim to quantify the contribution of each component and validate our design choices. Figure 3b illustrates the performance of various methods, including the aforementioned settings and some other fine-tuning strategies.

It is clear that both modeling the current layer strategy and propagate to the next layer strategy can work as a decent PEFT method alone, however, when used together, the improvement would be more complete with a higher expressivity.

### 5 CONCLUSION AND FUTURE WORK

The primary contribution of our paper is the introduction of the concept of propagating the feature adjustment value forward to the parameters of the next layer. This concept is motivated by Heterosynaptic Plasticity observed in BNN during LTP/D occurrences. We have conducted an analysis on the properties of this propagation, demonstrating its regularization abilities and how it enhances fine-grained expressivity through reparameterization perspectives. Moreover, we hypothesize that current feature tuning methods implicitly propagate, but by making this propagation explicit, we can simplify the learning process. Our experiments validate our concept, and we believe future work should focus on discovering how to propagate and re-apply additional parameters created for adjusting certain layer's features to more layers. By doing so, we can achieve even lower training costs and emphasize the interconnections between different layers.

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
