# OpenReview forum: "Discovering Long-Term Effects on Parameter Efficient Fine-tuning"
_ICLR.cc/2025/Conference — Submitted to ICLR 2025_

### Official Review · Reviewer_GuVH · 2024-10-16

**Soundness:** 3
**Presentation:** 2
**Contribution:** 2
**Rating:** 3
**Confidence:** 4

**Summary:**

Inspired the so-called heterosynaptic plasticity effect of biological neurons, the paper introduces a new parameter-efficient-fine-tuning method that fine-tunes per-feature scaling factors for pre-trained features. The key insight is that propagating the scales forward to the subsequent layer to mimic the effects known from classical neuroscience.

**Strengths:**

- The paper presents a method with clear inspiration from neuroscience.
- The mathematical explanation of the method is clear and extensive.
- The visuals of Figure 1 are appealing.

**Weaknesses:**

- While the visuals of Figure 1 are appealing, I did not understand what is shown in the bottom panel.
- Similarly, I did not understand Figure 2 at all.
- The evaluations are not clear. Implementation details are lacking. Please add the most important ones (tuning procedure, learning rates, ...) in the main paper and provide the full evaluation details via appendix or reference to related work. A hyperparameter tuning for all methods is essential (learning rate, rank, ...), as e.g. the paper considers LoRA only with rank=8 which is quite high for LoRA training and LoRA can be quite sensitive to the used rank. VTAB-1K provides a validation set on which hyperparameters should be tuned.
- A key motivation for PEFT is to fine-tune large foundation models. The used ViT-B model (pre-trained on ImageNet-21K) is extremely old as the reported ImageNet-1K full fine-tuning performance of 83.58% can be achieved while training on ImageNet-1K alone [1] with better protocols. Therefore, the trained models are far from state-of-the-art which limits comparability to old PEFT methods. Newer PEFT methods use DINOv2 (e.g. BOFT [2]) for vision tasks.
- Evaluation scope is limited to vision tasks. Recent PEFT methods use e.g. Language Generation (e.g. Dora [3]) as a benchmark for language models.
- As SAN is an improved variant of SSF, the evaluations should include comparisons against recent improvements of other PEFT methods (e.g. AdaLora [4] or PiSSA [5]).
- Modern ViT training protocols make heavy use of stochastic depth [6] which randomly drops blocks during training. This forces the model to not rely on the predictions of the previous block too much. As the proposed method propagates the learned scales forward between blocks, it could have vastly different behavior when stochastic depth is used. The considered ViT-B model does not use stochastic depth and is (as mentioned above) quite old. This point further shows the need for broader evaluation with current SOTA models like DINOv2 [7].
- Figure 3 should be tabulated somewhere (main paper or appendix).
- The title can be easily misinterpreted as the paper being an analysis of PEFT fine-tuned models and what their weaknesses are.

[1] Touvron et al, DeiT III: Revenge of the ViT - https://arxiv.org/abs/2204.07118

[2] Liu et al, Parameter-Efficient Orthogonal Finetuning via Butterfly Factorization - https://arxiv.org/abs/2311.06243

[3] Liu et al, Dora: Weight-decomposed low-rank adaptation - https://arxiv.org/abs/2402.09353

[4] Zhang et al, Adaptive budget allocation for parameter-efficient fine-tuning - https://arxiv.org/abs/2303.10512

[5] Meng et al, PiSSA: Principal Singular Values and Singular Vectors Adaptation of Large Language Models - https://arxiv.org/abs/2404.02948

[6] Huang et al, Deep Networks with Stochastic Depth - https://arxiv.org/abs/1603.09382

[7] Oquab et al, DINOv2: Learning Robust Visual Features without Supervision - https://arxiv.org/abs/2304.07193

**Questions:**

- Figure 1: What does "Restored" mean and how does it relate to neural networks? It would be nice to explain it such that readers not familar with biological neurons or neuroscience can understand it.
- Figure 2: How does linear probing fine-tune weights in the model (indicated by the flame)? What does positive and negative (red and blue) values refer to? What does "modeling" mean when referring a feature vector back to a weight matrix?
- Evaluation Details: Did you tune hyperparameters? What hyperparameter ranges did you use? Are the reported results averages across reruns?
- Scaling and scope: How does the method work on against scaled up foundation models like DINOv2 and on other tasks such as language generation or language understanding [1]?

Minor:
- Table 1 is hard to read because headers are only at the top, having method names in the intermediate gray areas would greatly improve it.
- The VTAB average results per category in Table 1 are swapped
- Citations are often "(Zaken et al., 2021)" instead of "Zaken (2021)". Examples: paragraph at line 77, 87, 153
- Paragraph at line 409 has weird sentence structure.
- Line 220: "VIT" should be "ViT"

[1] Wang et al, GLUE: A multi-task benchmark and analysis platform for natural language understanding - https://arxiv.org/abs/1804.07461

---

### Official Review · Reviewer_fvM6 · 2024-11-02

**Soundness:** 3
**Presentation:** 2
**Contribution:** 2
**Rating:** 5
**Confidence:** 5

**Summary:**

This paper proposes a lightweight parameter-efficient feature-tuning methodology inspired by Long-Term Potentiation (LTP) and Long-Term.
Depression (LTD) in biological brain. The method has been tested on various datasets with the methodology explained clearly and achieving relatively marginally higher mean accuracy.

**Strengths:**

1. A well-motivated paper and very interesting to read, especially those biological connections to LTP and LTD are intriguing.
2. Well-described previous methods and put the proposed algorithm in the state-of-the-art context.
3. All of the components of the method are well described, i.e. reparametrization, regularization, and explicit propagation.
4. Experiments are conducted on various datasets, with the proposed SAN achieving the highest accuracy.
5. Presenting a mathematical analogue of the LTP and LTD in section 3.2 is well explained and makes sense.

**Weaknesses:**

Despite the strengths, there are several weaknesses in the paper:

1. The proposed parametrization method is a simple extension to the scale-shift features (SSF) where only current layer parameters are modulated. The proposed method SAN also applies the SSF to the following layer parameter. Beyond that, I don't see a major contribution in methodology.
2. In order to justify the motivation of SAN, biological relation is presented multiple times, and it has consumed a lot of space in the paper. In my view, this paper needs a major rewrite where biological relations should be briefly described once in the paper. But it occurs repeatedly, which is unnecessary.
3. Related to Section 3.2.4, it is a redundant section and does not convey any information. It is simply a definition of SSF.
4. Table 1 shows mean accuracy on various small and bigger datasets such as ImageNet-1k. From the table, most of the focus has been set to mean accuracy; however, it is misleading and does not convey any strengths of the paper. A relatively higher mean accuracy across 5-6 datasets does not mean the method performs well across datasets.
5. The gains on imagenet-1k are marginal, which is the dataset of most interest.
6. The experiment section is quite weak. It must contain several other Transformer networks, such as EfficientVit and the latest Transformer architectures and their variants, because currently, only the bigger models are considered.

**Questions:**

1. Provide more experiments with different transformers, ConvNets and their variants, especially on ImageNet-1k and ImageNet-21k. This is needed to mark the importance of results in the paper.
2. I think biology is not needed much in the paper; hence, it is advisable to rewrite it. However, this would be a major rewrite.
3. Can you provide additional ablation studies on ImageNet-1k for different networks because it is the large datasets where performance becomes sensitive to the model design, in this case, the SAN?

---

### Official Review · Reviewer_NqRZ · 2024-11-03

**Soundness:** 3
**Presentation:** 4
**Contribution:** 4
**Rating:** 8
**Confidence:** 3

**Summary:**

This paper proposes a novel technique for Parameter Efficient Fine Tuning named SAN, which aims to learn a scaling factor for each layer of a pretrained network to determine its importance on the downstream task. The authors also show careful consideration as to the effect of this scaling factor on the subsequent layers. This technique uses a tiny fraction of the parameters that full finetuning would and also uses significantly less than other PEFT techniques while obtaining better end model performance on almost all datasets.

**Strengths:**

- This paper proposes a new PEFT technique which outperforms compared methods.
- The technique is biologically inspired and thus should be interpretable.
- The paper is well written and easy to follow.

**Weaknesses:**

- Although the paper focuses on parameter efficiency, what is the computational efficiency of the method compared to similar methods? This is in my opinion a very important factor for finetuning large pretrained networks. Maybe you could quantify this in terms of wall-time on the same PC or floating point operations?

**Questions:**

- Were there any particular architectural changes which had to be made in order to accommodate the swin/convnext models? Or is this method easy to drop in to any pre-trained model?
- I would appreciate if you would address my weakness.

---

### Official Review · Reviewer_KgNn · 2024-11-05

**Soundness:** 2
**Presentation:** 3
**Contribution:** 3
**Rating:** 3
**Confidence:** 4

**Summary:**

The paper introduces Synapses & Neurons (SAN), a novel fine-tuning method for Artificial Neural Networks (ANNs) inspired by the biological principles of Long-term Potentiation (LTP) and Long-term Depression (LTD). SAN draws an analogy between ANNs and Biological Neural Networks (BNNs), where ANN weights correspond to synapses and activations to neurotransmitters. By learning scaling matrices for features (activations) and propagating their effects across layers, SAN mimics the biological processes involved in learning and memory, allowing ANNs to acquire new knowledge more efficiently. The proposed method is evaluate on 26 datasets across attention-based and convolutional models and comparede with full finetuning, Visual Prompt Tuning, and Low-Rank Adapter methods by +8.5%, +7%, and +3.2%, respectively. This biologically inspired approach demonstrates SAN’s potential to enhance ANN adaptability and efficiency in fine-tuning tasks.

**Strengths:**

Writing and Motivation
- The paper is well motivated and well presented

Experiments and Method
- The experiments and comparison to baselines is quite thorough, except the focus of the paper is limited to computer vision applications
- The proposed finetuning scheme inspired from biological networks and its different properties are are studied thoroughly
- The proposed method is novel and outperforms PEFT methods like LoRA, BitFit, adapters, SSF on different computer vision tasks, while maintaining a smaller parameter count.

Significant and interest to the community
-The motivation of the method and results would be interesting to the community.

**Weaknesses:**

- Currently the study is limited to Vit-B and computer vision tasks. Given that the premise of the work if PEFT methods, I think the work would benefit a lot from scaling to billion parameter models, especially  Large language Models. The primary strength of PEFT methods is their scalability to very large models and this paper is currently missing studies on larger scales.
- Could the authors also compare to more recent PEFT methods like DoRA[1], Galore[2] and LoREFT[3]? Such baselines are missing from the work
- Currently I find the paper lacking ablations
         - Do you see the regularization effect mentioned in section 3.2.3, is this observed in practice?
         - Line 333 "Adaptive re-calibration of scaling factors" what is the effect of skipping the re-calibration step?
         - Is it important to propagate the scaling factors for all layers, or what is the effect of skipping propagation of the scaling factors for some layers?
         - Does the method transfer across model scales?

Minor Formatting: Line 199-200 are empty

[1]Liu, S.Y., Wang, C.Y., Yin, H., Molchanov, P., Wang, Y.C.F., Cheng, K.T. and Chen, M.H., 2024. Dora: Weight-decomposed low-rank adaptation. arXiv preprint arXiv:2402.09353.

[2]Zhao, J., Zhang, Z., Chen, B., Wang, Z., Anandkumar, A. and Tian, Y., 2024. Galore: Memory-efficient llm training by gradient low-rank projection. arXiv preprint arXiv:2403.03507.

[3]Wu, Z., Arora, A., Wang, Z., Geiger, A., Jurafsky, D., Manning, C.D. and Potts, C., 2024. Reft: Representation finetuning for language models. arXiv preprint arXiv:2404.03592.

**Questions:**

- Check weaknesses
- Could the authors elaborate on the hyperparameter settings for the method? Also are the initialisations of different learnable factors specific/tuned to a specific problem domain, and do the authors expect them to generalize well?

---

### Meta-Review · Area_Chair_dXLX · 2024-12-22

**Metareview:**

The submission proposes a parameter-efficient finetuning method for neural networks, termed Synapses & Neurons (SAN), inspired by biological neural networks.
During the initial round of reviews, the submission received scores of 3, 3, 5, 8. No rebuttal was provided by the authors. Weaknesses highlighted by the reviewers includes lack of sufficient experiments and ablations, as well as shortcomings in the writing.
The AC finds no reason to overturn the negative consensus of the reviewers.

**Additional Comments On Reviewer Discussion:**

No rebuttal provided by the authors.

---

### Decision · Program_Chairs · 2025-01-22

Reject